# Electro-assisted printing of soft hydrogels via controlled electrochemical reactions

Aruã Clayton Da Silva [1], Junzhi Wang[1] & Ivan Rusev Minev [1,2✉]

Hydrogels underpin many applications in tissue engineering, cell encapsulation, drug delivery and bioelectronics. Methods improving control over gelation mechanisms and patterning are still needed. Here we explore a less-known gelation approach relying on sequential electrochemical–chemical–chemical (ECC) reactions. An ionic species and/or molecule in solution is oxidised over a conductive surface at a specific electric potential. The oxidation generates an intermediate species that reacts with a macromolecule, forming a hydrogel at the electrode–electrolyte interface. We introduce potentiostatic control over this process, allowing the selection of gelation reactions and control of hydrogel growth rate. In chitosan and alginate systems, we demonstrate precipitation, covalent and ionic gelation mechanisms. The method can be applied in the polymerisation of hybrid systems consisting of more than one polymer. We demonstrate concomitant deposition of the conductive polymer Poly(3,4-ethylenedioxythiophene) (PEDOT) and alginate. Deposition of the hydrogels occurs in small droplets held between a conductive plate (working electrode, WE), a printing nozzle (counter electrode, CE) and a pseudoreference electrode (reference electrode, RE). We install this setup on a commercial 3D printer to demonstrate patterning of adherent hydrogels on gold and flexible ITO foils. Electro-assisted printing may contribute to the integration of well-defined hydrogels on hybrid electronic-hydrogel devices for bioelectronics applications.

[1] Department of Automatic Control and Systems Engineering, Faculty of Engineering, University of Sheffield, Sheffield, UK. [2] Leibniz Institute of Polymer Research Dresden, Dresden, Germany. ✉email: i.minev@sheffield.ac.uk

Hydrogels mimic many of the physical and biological properties of soft tissues[1–4]. As engineering materials, they offer a broad design space where viscoelasticity, swelling, bioactivity, degradation, space charge and other properties can be tuned. Applications in tissue engineering[5–7], drug delivery[8,9] or soft actuators[10,11] are already well known. More recently, hydrogels have attracted attention in the field of bioelectronic interfaces because the incorporation of conductive polymers can enhance their electrical properties. A number of hybrid materials containing PEDOT, PPy, PANI and a biomaterial formed from starch, PVA, acrylic acid, polyacrylamide, alginate, decellularised extracellular matrix or others have been reported[12–21]. Such materials can form the basis of technologies for interfacing living tissues as more biocompatible and versatile alternatives to metals and plastics.

Recent studies have reported use of conductive hydrogels in controlled drug delivery, as coatings on electrode contacts or entirely replacing the conductive materials in electrode arrays[22–27]. Similar work aiming to interface or regenerate cardiac tissues is also reported[28,29]. Typically, hydrogels are processed from an aqueous solution of monomers or macromolecules via polymerisation initiated by light, temperature or pH changes. They can be patterned by casting the pre-gel solution in moulds or for hydrogels that are already formed, direct ink writing (extrusion) or lithography may be employed. Despite these promising developments, the integration of well-defined hydrogel systems in devices remains a challenge. This is because hydrogel formation requires precise control over polymerisation reactions that determine the physical and biological properties of the material. Integration in useful devices is further complicated by the need to adhere the hydrogel to other materials such as metals or polymers.

A less explored route to gelation relies on aqueous electrochemistry. Electrochemical methods are a promising approach because they allow the formation of an adherent hydrogel on the surface of a metallic conductor (electrode) under benign conditions and without the need to add chemical catalysts. An early report on an electrochemically deposited hydrogel emerged in 2010 where electrolysis was used to promote abrupt pH changes close to an electrode-electrolyte interface as a gelation trigger for chitosan[30]. Later, the same group reported the electrodeposition of ionically crosslinked alginate hydrogel based on a similar principle[31]. Additionally, they proposed a covalently crosslinked chitosan hydrogel by using anodic oxidation of chloride (Cl⁻) anions[32]. A considerable number of other hydrogel systems based on this principle have been published since, among them layer-by-layer coatings[33,34] and chitosan/gelatine using pin art device[35].

Confining the electrochemical reaction volume to within small droplets may provide a convenient method to pattern the hydrogels into useful designs or integrate them directly on devices. Meniscus-confined 3D electrodeposition was proposed in 2010[36] as a method to control electrochemical reactions with good spatial resolution. The system was miniaturised using microelectrodes and a microreservoir adapted from a scanning electrochemical cell microscopy (SECCM) system[37,38]. Regarding hydrogel electrodeposition with spatial control, alginate and alginate/gelatine hydrogels have been patterned using a custom 3D printer configuration employing 2-electrodes[39,40]. While two electrodes are sufficient to trigger electrochemical reactions (e.g., electrolysis), selecting which reactions occur and controlling their rate is challenging because electrode potentials are uncertain. When producing biomaterials such as hydrogels, poor control over gelation reactions may have implications on the biodegradation, mechanical, electrical or biochemical properties of the material.

In this work, we describe the electrochemical formation of hydrogels confined in water droplets. Unlike previous approaches that rely on 2-electrodes[39,40], a crucial innovation is the introduction of potentiostatic control via 3-electrodes configuration. This enables gelation reactions to proceed at well-defined potentials and enables the formation of diverse hydrogel systems based on covalent or ionic crosslinking. As model systems, we used the polysaccharides chitosan and alginate. We further demonstrate polymerisation of a hybrid system where the conductive polymer PEDOT is polymerised alongside alginate. To demonstrate spatial control, we patterned shapes in 2D. We constructed a simple setup by linking a commercial 3D printer with a potentiostat. This was done by modification of standard printing nozzles to incorporate a (Ag/AgCl) pseudoreference electrode. Our electro-assisted printing approach may facilitate the integration of hydrogels in bioelectronic devices, such as electrode arrays or drug delivery systems with active coatings.

## Results

**Mechanism and setup for electro-assisted printing.** A 2-electrodes configuration offers a simple way of catalysing chemical reactions driven by local pH changes (Supplementary Fig. 1). Water hydrolysis causes depletion or increase of protons (H⁺) close to the cathode or anode respectively[31,32,41]. Although hydrogels can be produced in this way, the disadvantage of using 2-electrodes configuration is the poor kinetic control over electrochemical reactions. In 2-electrodes configuration, it is possible to apply constant current but the resulting electric potential will vary indiscriminately. An alternative is to introduce a third electrode and operate the system under potentiostat/galvanostatic control. Under these circumstances, the Nernst equation[42] and electrochemical kinetics[43,44] concepts are valid. Electrochemical reactions at the WE can then be easily identified and exploited.

Figure 1a shows a simplified overview of the hydrogel electro-assisted deposition process. Deposition follows an ECC mechanism[45,46], where typically an ionic species and/or molecule present in solution is oxidised over the conductive surface at a specific electric potential. The oxidation generates an intermediate species that reacts with the macromolecule, resulting in the formation of a hydrogel at the interface. In our printing setup, either a gold plate or ITO/PET conductive substrate is used as WE. As illustrated in Fig. 1b, the customised nozzle houses both the CE and the RE. The plastic casing of the nozzle prevents a short circuit between RE and CE. The setup enables not only droplet confined electro-assisted deposition but also material characterisation via electrochemical techniques such as CV, EIS and CA (Fig. 1c, top inset).

**Covalently crosslinked chitosan hydrogel.** As model hydrogel systems we used the polysaccharides chitosan and alginate because they are widely used in tissue engineering[47]. In the case of chitosan, the hydrogel can be either precipitated or covalently crosslinked[48] (Fig. 2a). The first step for both pathways is the dissolution of chitosan under acidic conditions (1% acetic acid). For precipitated gel, the second step is the deprotonation of chitosan which can be promoted by depleting the concentration of protons at the cathode by applying −2V (or more negative potential) to WE. This reaction can thus be driven by water hydrolysis. The resultant product is precipitated hydrated chitosan. An alternative chemical reaction can lead to the formation of covalently crosslinked chitosan hydrogels. This relies on the conversion of 4 chlorides to tetrachloroaurate (AuCl₄⁻), at the gold surface with gold being oxidised to gold (III)[49]. The characteristic peak for the tetrachloroaurate reaction occurs at 1.8 V. Afterwards, the primary alcohol group (-OH) of chitosan is

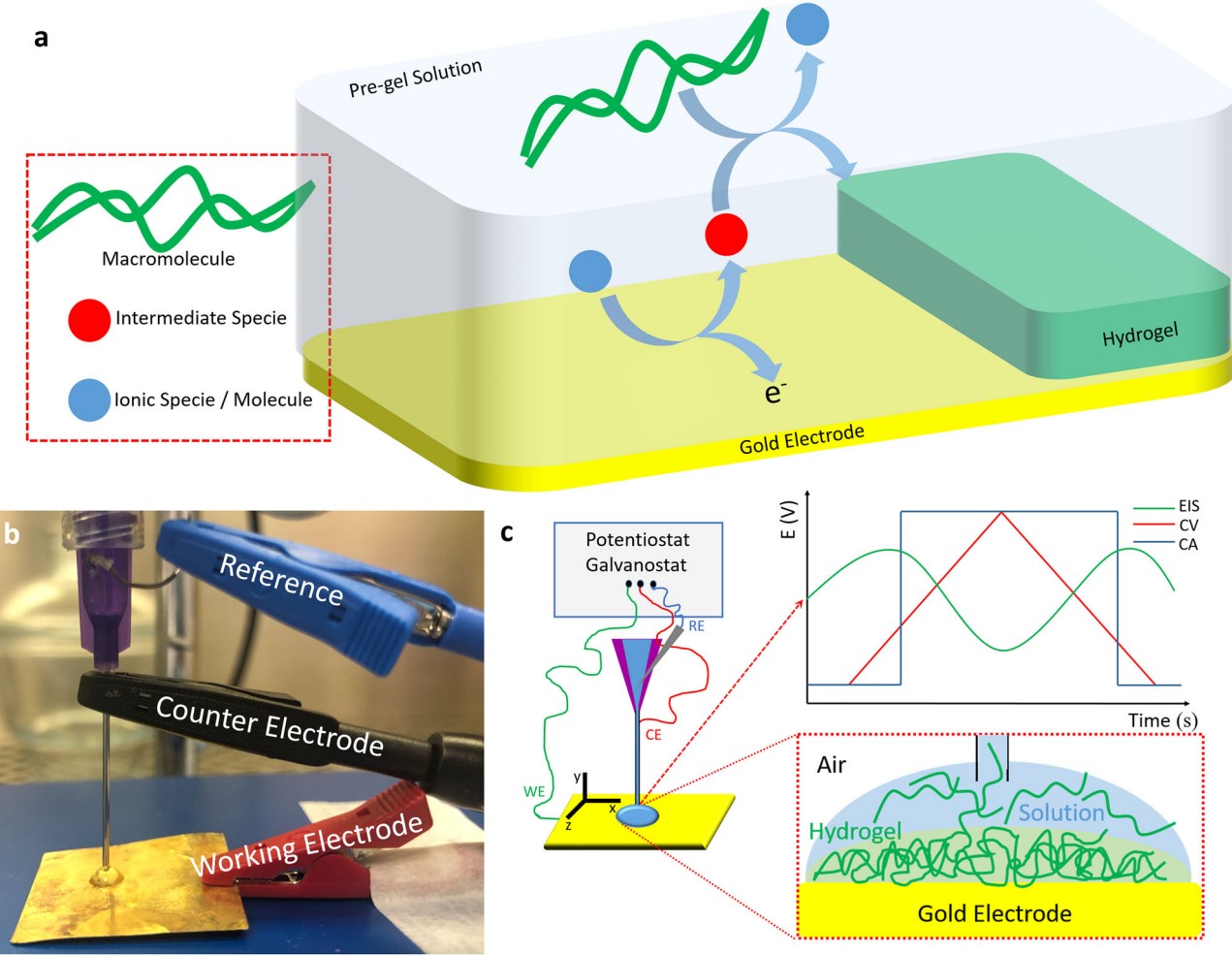

**Fig. 1 Electro-assisted hydrogel deposition and configuration of the electrochemical/3D-printing setup. a** Schematic representation of the general mechanism of electro-assisted hydrogel deposition over a conductive surface. An ionic species or molecule (blue sphere) is oxidised over the electrode (losing electrons), generating an intermediate species (red sphere) that reacts with the macromolecule (either chitosan or alginate, green) leading to crosslinking and deposition of the hydrogel. **b** Photograph of the 3-electrodes configuration with WE connected to a gold plate, needle of the customised nozzle connected as CE and Ag/AgCl pseudoreference wire connected as RE. **c** Schematic representation of the 3-electrodes configuration connected to a potentiostat/galvanostat enabling deposition and electrochemical measurements in situ. Insets: (above) example of electrochemical experiments (electrochemical impedance spectroscopy, EIS, green; cyclic voltammetry, CV, red and chronoamperometry, CA, blue), (below) schematic representation of hydrogel formation in a droplet.

oxidised to aldehyde group (-CHO) by tetrachloroaurate in solution. The final step also occurs in solution and consists of a Schiff base reaction where the amino group in chitosan covalently binds to the aldehyde's carbon forming a very stable double bond.

Previously it was reported that covalently crosslinked chitosan hydrogels are formed via hypochlorite ($ClO^-$) generated from oxidation of chloride to chlorine gas ($Cl_2$), followed by dissolution of the gas in aqueous solution[30,32,41]. However in our system we believe gelation proceeds via tetrachloroaurate generation instead. This is because we were unable to form covalently crosslinked chitosan in the absence of a gold surface (i.e. on an ITO surface). Furthermore, if the electrochemical reaction generates gas (in this case chlorine species), we would expect to see small cavities at the bottom of the hydrogel and perturbations in the CV current near the reaction peak, when gas bubbles change the electrode surface in contact with solution (Supplementary Fig. 2).

Figure 2b depicts chitosan hydrogels deposited at −2 V and at 1.8 V (vs Ag/AgCl pseudoreference). Following repeated washing in deionized water, both types of hydrogel remain attached to the

WE surface however, they have different appearance. The chitosan deposited at −2 V is cloudy and contains many cavities (Fig. 2b-i). We assume that these hydrogels are formed according to the precipitation reaction in Fig. 2a. Hydrogels deposited at 1.8 V on the other hand are clear (Fig. 2b-ii). Conducting the deposition at the tetrachloroaurate peak (1.8 V) is expected to favour the deposition of covalently crosslinked gels because of the chloride reaction generating tetrachloroaurate. In the absence of chloride, only a much smaller peak is present due to gold going to gold(I). We attempted to deposit chitosan in the absence of chloride ions (at 1.8 V, red trace in Fig. 2d), but only a small amount of precipitated chitosan was observed on the CE, likely due to water hydrolysis (Supplementary Fig. 3).

Figure 2c shows lateral views of the deposition of covalently crosslinked chitosan hydrogels. The hydrogel growth during deposition of low and high $M_w$ chitosan is quantified in Fig. 2e, f respectively. It is noted that the hydrogel thickness reaches a plateau. Likely this is because thicker gels present an obstacle for oxidative species to diffuse away from the gold surface. As the gel

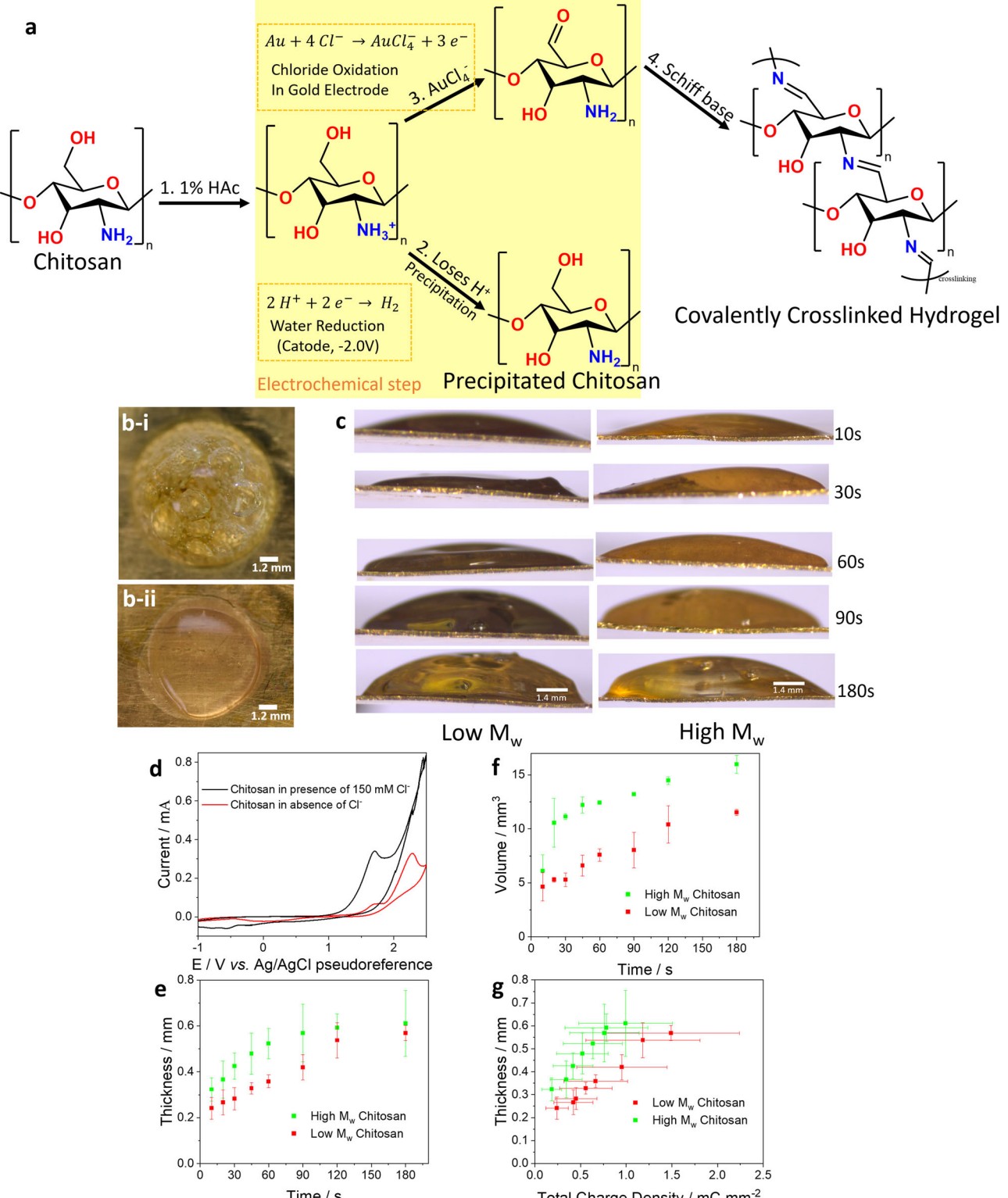

**Fig. 2 Reactions and growth profiles for chitosan hydrogels. a** Schematic representation of the chemical reactions involved in chitosan hydrogel formation, yellow block highlights the electrochemical controlled steps that can lead to precipitated (by water hydrolysis) or covalently crosslinked (gold oxidation in presence of chloride) hydrogels. **b** Representative picture of the (**i**) precipitated (−2.0 V) and (**ii**) covalently crosslinked (+1.8 V) high molecular weight ($M_w$) chitosan. **c** Representative pictures (side view) of the growth profile of low and high $M_w$ chitosan. **d** CV of chitosan in presence of 150 mmol L$^{-1}$ (black) and absence (red) of chloride in the supporting electrolyte. Chloride was substituted by sulfate. Scan rate was 100 mV s$^{-1}$. **e** Thickness and **f** volume growth of high (green) and low (red) $M_w$ chitosan as a function of polymerisation. The applied potential is +1.8 V vs Ag/AgCl. **g** Relationship between thickness and total charge density supplied to promote the gelation reaction for high (green) and low (red) $M_w$ chitosan. Thickness, volume and total charge density here are measured quantities with the error bars representing the mean ± standard deviation ($n = 3$ independent electrodeposition experiments).

thickness increases, its growth kinetics is slowed down because freshly generated oxidative species need to diffuse through thicker chitosan layers before reaching free macromolecules at the outer interface. In addition, by passing through the formed gel, tetrachloroaurate may oxidise other alcohol groups inside the hydrogel matrix. The 3-electrodes setup allows us to precisely measure the current during hydrogel formation via CA. This allow us to relate charge to the thickness of hydrogels (Fig. 2g). If we assume that the initial growth phase is linear (thickness $< 0.5$ mm), we obtain growth profiles of 0.83 ($r = 0.9930$) and 0.52 mm$^3$ mC$^{-1}$ ($r = 0.9931$) for the high and low $M_w$ chitosan, respectively. This means that more hydrogel is formed per unit charge for high $M_w$ chitosan (1.6 times faster) compared to low $M_w$. This is likely because fewer macromolecules (bonds) are required to produce the same volume of material.

**Ionically crosslinked alginate hydrogel**. Potentiostatic (3-electrodes) electro-assisted deposition can be useful for improving growth control of ionically crosslinked gels. In our second model system, we used controlled water hydrolysis to deposit alginate hydrogels. Although the reaction here is different than for chitosan, the formation of ionically crosslinked alginate hydrogel also follows an ECC mechanism. Figure 3a shows the chemical reaction of calcium ($Ca^{2+}$) complexation with carboxylic groups in alginate macromolecules. The electrochemical step is related to production of protons at the WE interface (using water hydrolysis) to promote the decomposition of calcium carbonate. This releases calcium for the complexation reaction. Figure 3b shows representative pictures of the growth profile of alginate hydrogels under different WE potentials of 3 V, 4 V and 5 V. Figure 3c shows a CV of the gold WE in alginate solution. Sharp current increase beyond 2.5 V (*vs* Ag/AgCl pseudoreference) is associated with water hydrolysis and proton generation at the WE. Figure 3d, e quantify the growth profile in terms of thickness and volume ($n = 3$). It suggests that hydrogel growth is faster at higher potentials. This can be explained by higher energy promoting higher rates of proton production. Beyond a thickness of approximately 0.6 mm gel growth slows down which can be attributed to mass transport limitation and electrical passivation effects of the already formed hydrogel at the interface. The hydrogel thickness appears to be proportional to the cumulative amount of supplied charge density (Fig. 3f). The growth profiles (thickness per unit of charge) for gels grown at 3, 4 and 5 V are not significantly different. An average growth profile of 0.17 mm$^3$ mC$^{-1}$ was extracted. This result indicates that at higher potentials, the rate of proton generation is enhanced but the gelation mechanism remains the same. The growth profile of alginate hydrogel is 4.8 and 3 times smaller than high and low $M_w$ chitosan hydrogels, respectively. Due to the different gelation mechanisms, a direct comparison between the growth profiles of chitosan and alginate is challenging.

**Hybrid PEDOT/alginate conductive hydrogel**. In our third model system, we attempted to promote two polymer systems simultaneously. We studied the electrodeposition of PEDOT together with alginate. Figure 4a illustrates the electro-polymerisation reaction of EDOT to PEDOT. During the polymerisation of PEDOT excess positive charges are compensated by negative charges (COO$^-$) on alginate macromolecules. Thus we expect to produce PEDOT doped with alginate while simultaneously, the proton generated during the PEDOT electrodeposition supports the gelation process of alginate (without the need for water hydrolysis), resulting in a hybrid PEDOT/alginate hydrogel. The resulting polymerisation process by applying $+1.8$ V (*vs* Ag/AgCl pseudoreference) is shown in Fig. 4b. Over a

time of several minutes, the droplet changes colour from translucent to dark blue. This colour change is indicative of a successful PEDOT polymerisation. PEDOT polymerisation appears to start directly below the needle position and extends radially until the whole gold surface delimited by the droplet is covered in PEDOT (Supplementary Movie 1). Beyond 6 min, as the electrodeposited material covered the WE surface, the polymerisation current (Fig. 4c) begins to decrease which is likely due to depletion of monomers in the electrolyte. Figure 4d shows CV of the alginate solution with addition of 50 mM EDOT. In the first cycle, no significant peaks are observed (clear surface of gold electrode). However, at $+1.7$ V (*vs* Ag/AgCl pseudoreference) starts an increasing current correspondent to the EDOT electropolymerisation. After several cycles a peak around $+1.25$ V (*vs* Ag/AgCl pseudoreference) emerges which is related to the polymerisation of free EDOT monomers over already formed PEDOT surface[50]. In addition, two broad peaks characteristic of PEDOT oxidation and reduction ($+0.35$ V and $-0.45$ V, respectively) also appear. These observations suggest formation of a PEDOT/alginate hydrogel in the droplet. To trigger the formation of this hybrid hydrogel we applied $+1.8$ V (*vs* Ag/AgCl pseudoreference) to the WE as this is the potential at which the PEDOT polymerisation reaction starts but water hydrolysis is avoided.

In typical electrochemical cells, the area of the CE is considerably larger than that of the WE. In our setup, the area of the CE comprises not only the tip of the nozzle but also its internal walls. With an estimated area of 40.7 mm$^2$ the CE in our experiments is nearly two times larger than the footprint of alginate and chitosan droplets. Accordingly, we did not observe any side reaction or obstruction of the nozzle during gelation.

In addition to electrodeposition, our 3-electrodes setup allows us to perform EIS to study the electrical properties of the various gels. It is worth mentioning that the electrochemical experiments were made in situ, in the polymerisation solution itself. Impedance spectra of the alginate containing gels systems is presented in Fig. 4e. The electrical behaviour of gels was modelled on the modified Randles equivalent circuit model (Fig. 4f) which allowed us to extract and compare values for the putative electronic elements of resistance and capacitance (Fig. 4g). We observed that the alginate hydrogel resistance (0.83, 0.94 and 4.72 Ω m$^2$) and capacitance (23.74, 48.31 and 26.10 μF) increase with the increase in deposition potential (3, 4 and 5 V, respectively). This may be due to the formation thicker denser gels with better insulating properties and higher space charge close to the gold surface. When PEDOT is polymerised together with alginate a similar resistance of alginate 3 V (0.84 Ω m$^2$) as well as a considerable increase (2 to 4 times) in capacitance (101 μF) were observed. Additionally, for the hybrid PEDOT/alginate we used a modified equivalent circuit to discriminate the PEDOT and alginate contributions (Supplementary Fig. 4). The PEDOT contribution presented significantly smaller resistance (0.20 Ω m$^2$) and capacitance (0.54 μF). It clearly suggests the addition of conducting polymer enhanced the conductivity and charge transfer of the material.

Our findings suggest that differences in the kinetics and stochiometry of the ECC mechanisms influence the rate of growth of the hydrogel systems. The chitosan and alginate hydrogels deposit fast and homogeneously throughout the conductive surface. A chitosan or alginate hydrogel covering the footprint of the droplet can be formed in as little as 10 s. Probably because of rapid electron transfer from ions (chloride) or water molecules to the WE (showing as high current peaks in CV), followed by fast chemical (crosslinking) steps. For the chitosan and alginate systems the electrochemical reaction generates cross-links between two macromolecules. In contrast electrodeposition of PEDOT proceeds by chain formation from

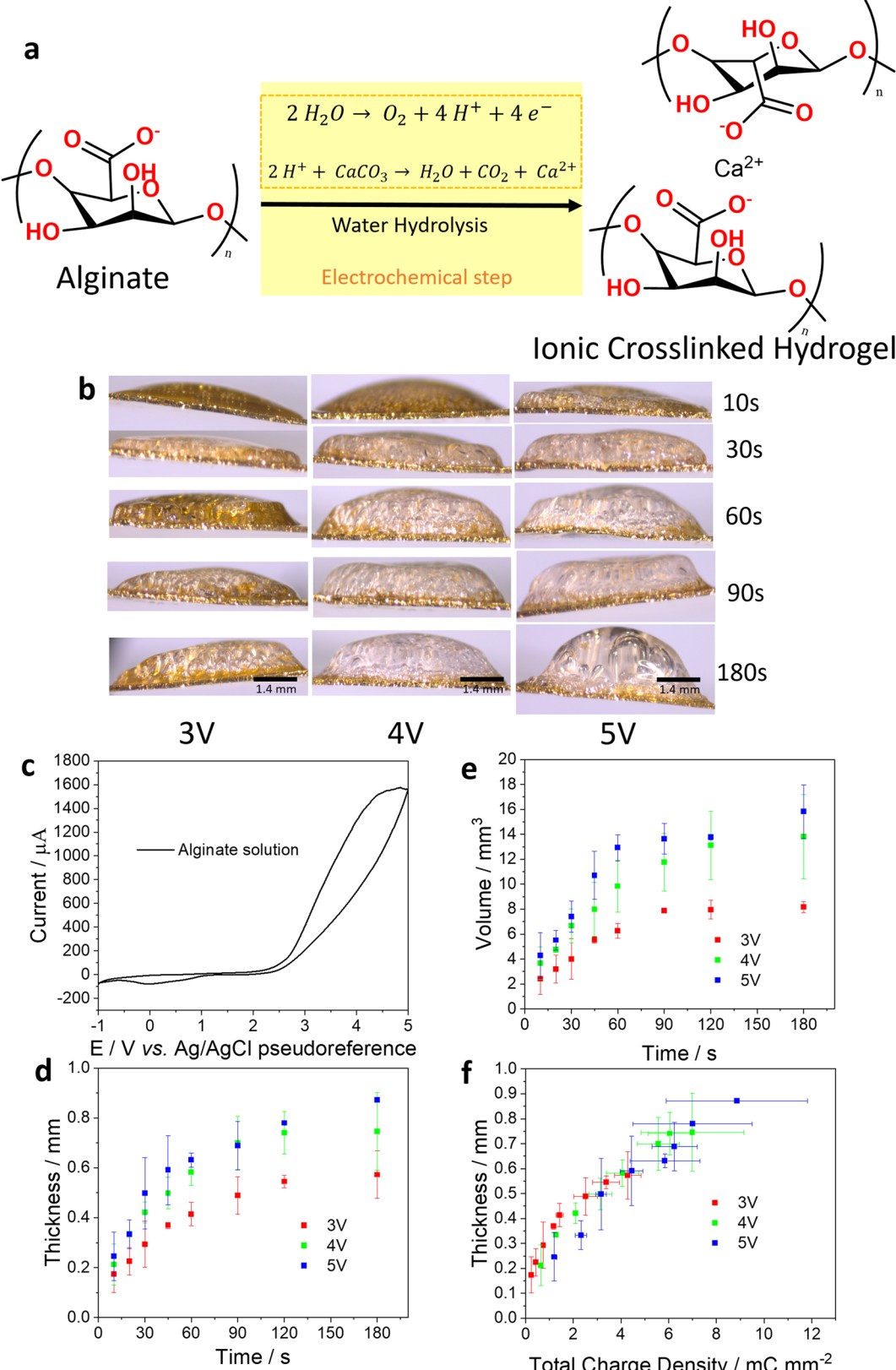

small EDOT monomers. This is consistent with our observation that the PEDOT/alginate hybrid forms at a much slower rate. The electrodeposition rate may be further limited by slow diffusion of EDOT monomers in the viscous alginate solution. A PEDOT/alginate hydrogel covering the available WE area was observed to take approximately 6 min.

**Electro-assisted printing hydrogels.** Finally, we explored approaches to pattern hydrogels in 2D. In the experiments above the gelation volume is a droplet bound by the gold plate and the dispensing needle. By introducing relative movement between them during polymerisation, we printed hydrogel lines of arbitrary design. This depends on a careful selection of nozzle speed

**Fig. 3 Reactions and growth profiles for alginate hydrogels. a** Schematic representation of the chemical reactions involved in the deposition of ionically crosslinked alginate hydrogels. The yellow block highlights the electrochemically controlled steps that generate protons and promote the gelation reaction. **b** Representative pictures (side view) of the growth profile of alginate hydrogels electrodeposited at 3, 4 and 5 V, respectively. **c** CV of alginate solution as supporting electrolyte at scan rate of 100 mV s⁻¹. **d** Thickness and **e** volume of 3 V (red), 4 V (green) and 5 V (blue) as a function of gelation time. **f** Relationship between thickness and total charge applied with 3 V (red), 4 V (green) and 5 V (blue) to promote the gelation reaction. Thickness, volume and total charge density here are measured quantities with the error bars representing the mean ± standard deviation ($n = 3$ independent electrodeposition experiments).

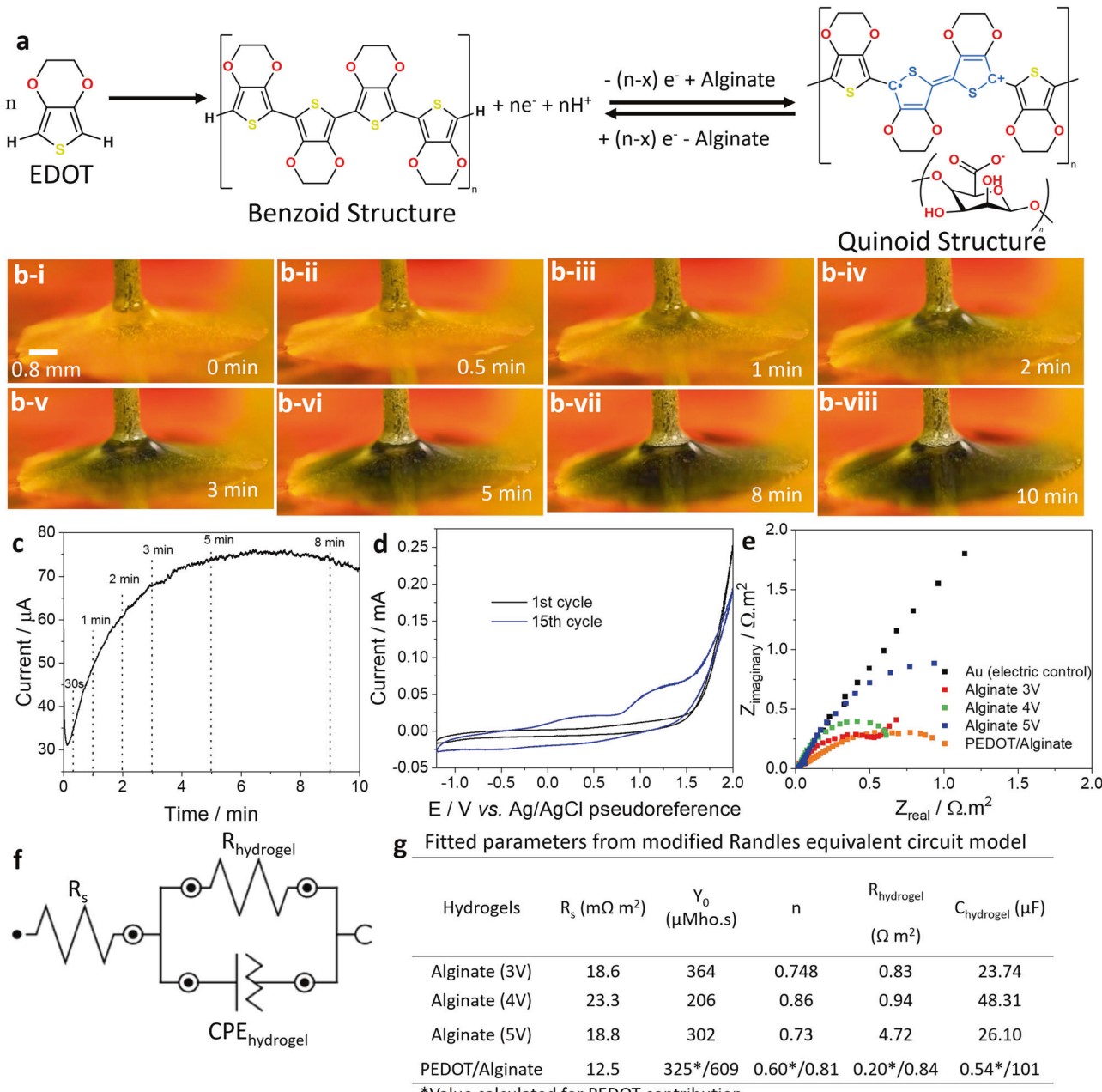

**g Fitted parameters from modified Randles equivalent circuit model**

| Hydrogels | $R_s$ (mΩ m²) | $Y_0$ (μMho.s) | n | $R_{hydrogel}$ (Ω m²) | $C_{hydrogel}$ (μF) |
|---|---|---|---|---|---|
| Alginate (3V) | 18.6 | 364 | 0.748 | 0.83 | 23.74 |
| Alginate (4V) | 23.3 | 206 | 0.86 | 0.94 | 48.31 |
| Alginate (5V) | 18.8 | 302 | 0.73 | 4.72 | 26.10 |
| PEDOT/Alginate | 12.5 | 325*/609 | 0.60*/0.81 | 0.20*/0.84 | 0.54*/101 |

*Value calculated for PEDOT contribution.

**Fig. 4 Hybrid PEDOT/alginate hydrogels and electrochemical characterisation. a** Schematic representation of the chemical reactions of EDOT polymerisation (left) and redox changes with alginate as dopant between benzoid and quinoid (blue) structures of PEDOT (right). **b** Selected frames of PEDOT/alginate electrodeposition. Scale bar 0.8 mm. **c** CA obtained during PEDOT/alginate polymerisation at +1.8 V with selected times in dashed line. **d** 1st (black) and 15th (blue) CV cycles in alginate solution with 50 mM EDOT at scan rate of 100 mV s⁻¹. **e** EIS showing Nyquist plot of bare gold surface (black), alginate deposited at 3 V (red), 4 V (green), 5 V (blue) and PEDOT/alginate (orange). The alginate hydrogels were deposited in 60 seconds, while PEDOT/alginate hydrogel took 6 min. **f** Modified Randles equivalent circuit used to model the electrical properties of hydrogels, $R_s$ indicates solution resistance, $R_{hydrogel}$ is the hydrogel resistance and $CPE_{hydrogel}$ is the constant phase element for the hydrogel. **g** Equivalent circuit parameters extracted from impedance spectra, where $Y_0$ is the admittance, n is the deviation from ideal capacitive behaviour and $C_{hydrogel}$ is the capacitance of the hydrogel. Asterisk means the PEDOT contribution to the total value.

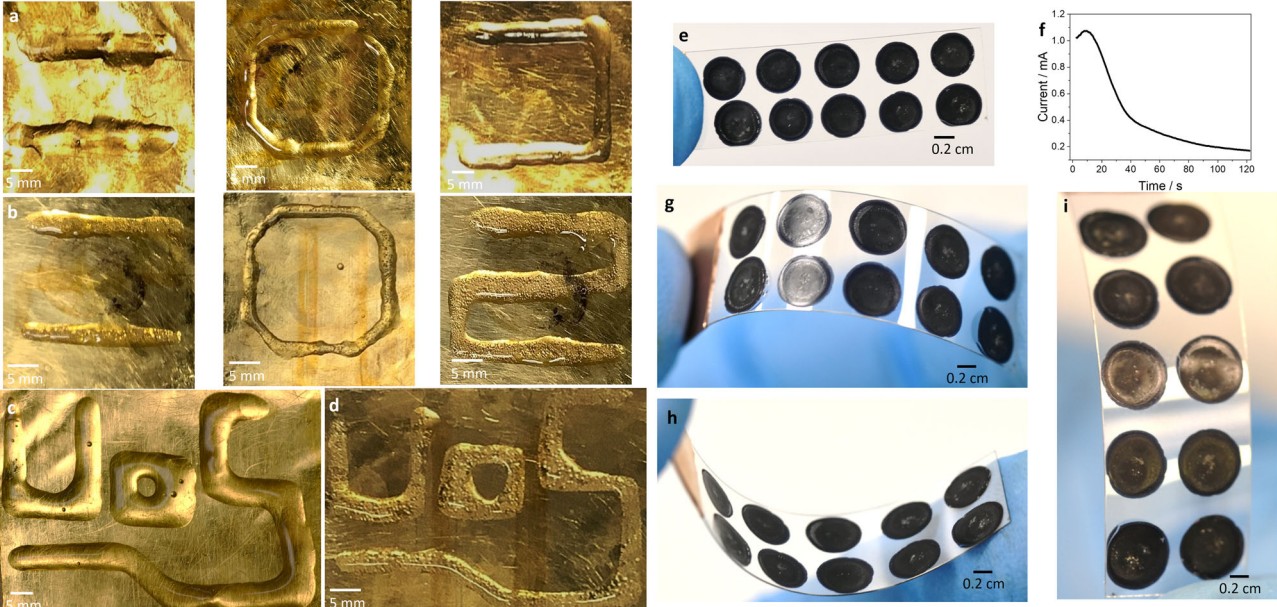

**Fig. 5 Electro-assisted hydrogel patterning. a** Representative pictures of deposition of different patterns with high $M_w$ chitosan (1.8 V) and **b** alginate (5 V) hydrogels, in lines (left), octagons (middle) and curved paths (right). **c** Increasing the complexity, the UoS pattern was printed using the high $M_w$ chitosan (1.8 V) and **d** alginate (5 V) hydrogels. **e** PEDOT/alginate hydrogel dots patterned on a rectangular piece (4.0 × 1.0 cm²) of ITO/PET flexible substrate. The polymerisation potential is 3 V. **f** Representative CA profile during deposition of the dots. **g–i** convex, concave and top view of the substrate with dots under bending, respectively.

(0.01 mm s⁻¹), rate of fresh pre-gel solution supply (0.25 µl s⁻¹) and the electrochemical parameters. It is also critically important to control the distance between plate and needle in order to avoid open circuits (needle high) or short circuits (needle touching). Keeping this distance constant during a print run ensures the droplet maintains a constant footprint on the gold plate. In our setup, the movement of the customised printing nozzle and the supply of pre-gel solution were controlled by a commercially available 3D printer (3DDiscovery, RegenHU) equipped with a volumetric dispenser. In Fig. 5, we illustrate printing of the various hydrogel systems. For all printing experiments, the nozzle was equipped with a needle with 510 µm inner diameter which results in line widths of 5.7 and 4.5 mm respectively for chitosan and alginate hydrogels when printing over gold without any surface modification (Fig. 5a, b). As a demonstration, the letters UoS were printed using chitosan and alginate (Fig. 5c, d). Within the confines of the gold plate WE (2.5 × 2.5 cm²) no significant differences in the electrochemical control were observed. The printing method is versatile and can be applied to substrates used in flexible electronics such as ITO on PET. We first experimented with the alginate system (Supplementary Fig. 5). At 5 V, deposition on the ITO/PET substrate was successful, however there are some limitations. The water hydrolysis reaction used to generate alginate hydrogel strips the ITO layer (a thin oxide layer) due to gas generation. Nevertheless, alginate hydrogel deposition runs for approximately 2 min after which the current drops to zero due to loss of substrate conductivity. Afterwards, we electrodeposited PEDOT/alginate droplets on ITO/PET at in an array of dots (Fig. 5e). In this experiment, we applied +3 V vs Ag/AgCl pseudoreference to improve the kinetics of the polymerisation reaction. It is important to mention that in the PEDOT/alginate system, the deposition does not damage the thin ITO layer, because the water hydrolysis reaction is shifted to more positive energy. In this way, it is possible to apply an overpotential to promote faster electrodeposition without water hydrolysis destroying the WE or the PEDOT/alginate hydrogel. Figure 5f shows a representative CA profile during PEDOT/alginate dot

formation. The average charge for the electrodeposition process was 45.8 ± 5.3 mC ($n = 10$). The PEDOT/alginate dots remain attached to the substrate when it is flexed (Fig. 5g, h, i), demonstrating good adhesion to the substrate. Continuous lines of PEDOT/alginate can also be produced using this method (Supplementary Fig. 6). Droplet sizes and printed line widths are dependent on the needle diameter, viscosity and surface tension of the pre-gel solution as well as the surface energy of the printing substrate. Optimisation of line widths and printing parameters however is beyond the scope of this work.

## Discussion

In this work we describe electrochemically controlled polymerisation of ionic, covalent and hybrid hydrogels. Introduction of potentiostatic control (3-electrodes configuration) ensures polymerisation reactions can be selected and controlled to produce materials with defined properties. Since gelation occurs in spatially confined water droplets, the method is easily extended to patterning gels in 2D and is compatible with flexible conductive substrates. Our electro-assisted printing method contributes to a growing toolbox for design and processing of functional hydrogels with applications in biointerfaces and soft electronics.

## Methods

**Materials.** The chemical reagents, low (50–190 kDa) and high (310–375 kDa) molecular weight ($M_w$) chitosan (75–85% deacetylated), sodium chloride (NaCl), sodium alginate, calcium carbonate, sodium dodecyl sulfate (SDS) and 3,4-ethylenedioxythiophene (EDOT) were purchased from Sigma-Aldrich. Acetic acid (HAc, 99.7%) was purchased from Acros Organics. All solutions were prepared with deionized Milli-Q water (18.2 MΩ).

**Pre-gel solutions.** Chitosan hydrogels were prepared from solution containing 1% (m/v) chitosan (low or high $M_w$) dissolved in 1% (v/v) acetic acid with addition of 0.15 mol L⁻¹ of sodium chloride under room temperature and magnetic stirring for 1 h. Alginate hydrogels were prepared from suspension containing 1% (m/v) sodium alginate dissolved in deionized water and suspended 0.5% (m/v) of calcium carbonate under room temperature and magnetic stirring for 1 h. PEDOT/alginate hydrogels were prepared using the same composition for alginate hydrogel with addition of 70 mM of SDS (to increase the solubility of EDOT[51]) and 50 mM of

EDOT. Pre-gel stock solutions were placed into syringes of 10 mL for subsequent electrochemical experiments.

**Electrochemical measurements in situ**. Cyclic voltammetry (CV), chronoamperometry (CA) and electrochemical impedance spectroscopy (EIS) were performed using a portable potentiostat (PalmSens4 controlled using PSTrace 5.8 software) coupled to the 3D printer. In-droplet EIS were recorded from 1 MHz to 0.1 Hz, with excitation amplitude of 10 mV (RMS) at 10 points per decade. The WE was either gold foil 0.10 mm thick, $2.0 \times 2.0$ cm$^2$ (Goodfellow, 99.95% purity) or indium-tin oxide coated polyethylene terephthalate (ITO/PET, resistivity 60 $\Omega$/sq, Sigma-Aldrich) cut in rectangular shape $4.0 \times 1.0$ cm$^2$. The WE was the stainless steel dispensing needle (510 μm inner diameter, 820 μm outer diameter and 25.4 mm length) of a commercially available printing nozzle (Intertronics, FIS5601099). The RE was prepared using silver wire 0.10 mm diameter, 2 cm length (Goodfellow, 99.99% purity) covered with electrodeposited silver chloride (AgCl) according to published protocols[52]. Briefly, the silver wire electrode is immersed in potassium chloride 3 mol L$^{-1}$ solution and open circuit potential is measured for 300 seconds. Later, $+50$ mV above the open circuit potential is applied for 1800 seconds. The RE was inserted in the printing nozzle and secured in place by partially melting its plastic casing and further fixed with cyanoacrylate glue. The RE is immersed in the pre-gel solution, however it does not make electrical contact with the needle CE. This assembly is herein referred to as a customised printing nozzle.

Total charge density (C.mm$^{-2}$) was calculated by $Q = I \times t$, where I is the current density (A.mm$^{-2}$) and $t$ is time (seconds). We applied the modified Randles equivalent circuit model to extract values for the constant phase element (CPE) and resistivity of hydrogels. Parameter fitting was conducted using the NOVA 2.1 software (Metrohm Autolab). Capacitance was calculated from the constant phase element (CPE) using $C_{hydrogel} = \frac{\sqrt[n]{R_{hydrogel} \times Y_0}}{R_{hydrogel}}$, where $C_{hydrogel}$ is capacitance of the hydrogel (Farad), $R_{hydrogel}$ is the resistance of the hydrogel ($\Omega$.m$^2$), $Y_0$ is the admittance (Mho or Siemens) and n is the deviation from ideal capacitive behaviour[53].

**Electro-assisted printing procedure**. The customised printing nozzle was attached to a syringe containing pre-polymer solution and mounted on the bioprinter 3DDiscovery (RegenHU, Switzerland). The WE (gold plate or ITO/PET) was laid flat on the printing platform and a droplet of pre-gel solution was dispensed on the surface. The customised nozzle was brought above the plate in contact with the droplet. For printing of complex shapes, pre-gel solution was continuously dispensed through the dispensing needle. Patterns of different design were generated in G-code with the software BioCAD. Following deposition hydrogels were gently rinsed with deionized water to remove any macromolecule, unpolymerised monomers or reaction byproducts. The thickness, area and volume of the generated hydrogels were obtained from optical micrographs (Stemi 508 Compact Greenough Stereo Microscope, Zeiss) with ImageJ software using spherical segment from top view and cylinder from lateral view. All experiments were made in triplicate using freshly prepared solutions. Unless stated otherwise, data is reported as the mean ± standard deviation.

## Data availability
The data that support the findings of this study are available from the corresponding author upon request.

## Code availability
The G-code for the 3D-printing patterns are available from the corresponding author upon request.

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

## Acknowledgements
All authors acknowledge funding from ERC Starting Grant: IntegraBrain (804005).

## Author contributions
A.C.D.S. and I.R.M. conceived the idea. A.C.D.S. developed the electro-assisted printing. A.C.D.S. and J.W. planned and performed the experiments. A.C.D.S. and I.R.M. analysed results and wrote the manuscript.

## Competing interests
The authors declare no competing interests.
