## [Peer Review File · Nature Communications]

REVIEWER COMMENTS

Reviewer #1 (Remarks to the Author):

This paper concerns electrochemical deposition of soft hydrogels and printing of the hybrid PEDOT / alginate gel with potentiostatic control and 3D printer.

The result of comparing the amount of electricity and the thickness of the gel is interesting. By comparing Fig. 2d and Fig. 3f, the difference in the formation rate of chitosan gel and alginate gel can be discussed. Please supplement the explanation about the error bar. Also discuss the formation rate of hybrid PEDOT / alginate gel.

Impedance (EIS)-based gel characterization is very significant (Fig. 4e & g). The unit in the text (and Methods section) is incorrect. $\text{k}\Omega \text{ (ohm) cm}^{-2}$ should be $\text{k}\Omega \text{ (ohm) cm}^{+2}$

P.8, Line 137; "However, the reported reaction occurs at +1.396V, while water hydrolysis oxidation is +1.230V, being impossible to obtain a clear hydrogel without cavities in aqueous media (Supplementary Fig. 2)." It is not sufficient here from the point of the difference in standard electrode potential alone. The reaction rates of equation a, b, and c may depend on the material of the electrode. The reaction rate may change depending on whether Cl^- is adsorbed on the electrode or not. It is recommended to consider at least multiple conditions for Cl^- concentration and perform electrochemical measurement. On the other hand, fig.2d compares only with and without Cl^- . Author should show the exact concentration of Cl^- ion for the fig.2d, with Cl^- ion condition.

Reviewer #2 (Remarks to the Author):

The current by Da Silva et al. describes an electro-assisted printing approach using a commercial 3D printer with a potentiostat. This was done by modification of standard printing nozzles to incorporate a (Ag/AgCl) pseudoreference electrode, using in this way a three-electrode set up.

The manuscript reads well with a well-executed experimental design. The formation of covalently formed hydrogels by the 3-electrode set up is quite interesting, and represents a step-change in capability compared to the 2-electrode systems used previously. The work will therefore be of

significance to those working in the field of soft neural electrodes where the connection between metal conductors and the polymer material can easily be compromised.

The methodology is sound and appears to be described in enough detail to undertake replication of the work.

This manuscript has the potential to show that the author's electro-assisted printing approach could offer an all-in-one set-up for controlled polymerization of hydrogels with patterning capabilities. My only suggestion is that the authors try to place a few more lines of text into the introduction to further emphasise the novelty of this work over the 2D electrode work that they introduce. Otherwise, I think this work is well suited for publication in Nature Communications.

Minor change:

References should be added for this work described as: "...previous approaches that rely on 2-electrodes"

Reviewer #3 (Remarks to the Author):

Paper Review

General Comments

The manuscript titled "Electro-assisted printing of soft hydrogels via controlled electrochemical reactions" describes a study using electrochemical processes to polymerise hydrogel materials using typical materials (alginate, chitosan), and hybrid materials that also incorporate the organic conducting polymer PEDOT. The study describes and characterises the use of electrochemical control to drive the polymerisation processes of the material, incorporating with this system with a fabrication approach that allows the patterning of the hydrogel material. This is an interesting study that presents an innovative approach to polymerising and patterning hydrogel-based materials for biomedical applications. I have some specific queries/comments for the authors consideration below.

Specific Comments

Lines 35 - 37. This sentence is confusing and is not clear. Please revise.

Lines 35 - 37. There are quite a number of self-citations from the authors when discussing the incorporation of conducting polymers in hydrogels and other types of biomaterials. However, there

are also other worthy seminal works that should be included. Please include some of the major works from the field that have been omitted from the references here.

Lines 100 – 108. The relative surface area of the counter electrode vs the working electrode will change depending on the diameter of the droplet on the surface. Did the authors identify any critical relationship between the ratio of CE:WE size and how this influenced the electrochemistry? This is particularly relevant for the characterisation of the materials using CV or EIS, as the surface area of the porous hydrogel material will (presumably) be significantly greater than the surface area of the needle counter electrode.

Also, given polymerisation can take some time (several minutes), did the authors have any issues with the needle becoming blocked through the polymerisation of material within the needle itself?

Line 143. Correct spelling to 'stabilized'.

Line 221 – 223. “Beyond 6 minutes, the polymerisation current begins to decrease probably because the electrodeposited material has covered the WE surface and starts to grow in thickness.” During this polymerisation process is material being actively released through the needle, or is it stagnant? If the flow of material is stagnant, the dropping off of the potential may be that the concentration of monomer in the polymerisation medium is becoming exhausted, and is not being replenished through introduction of new material through the needle, thus polymerisation rate is decreasing.

Line 224. Add bracket.

Reply to comments of reviewers:

We would like to thank the Editors and the reviewers for the careful reading of the Manuscript. Changes in the manuscript are highlighted in blue font.

Reviewer #1 (Remarks to the Author):

This paper concerns electrochemical deposition of soft hydrogels and printing of the hybrid PEDOT / alginate gel with potentiostatic control and 3D printer. The result of comparing the amount of electricity and the thickness of the gel is interesting. By comparing Fig. 2d and Fig. 3f, the difference in the formation rate of chitosan gel and alginate gel can be discussed. Please supplement the explanation about the error bar. Also discuss the formation rate of hybrid PEDOT / alginate gel.

We thank the reviewer for the positive evaluation of our work and prompting us to discuss several important points further.

The growth rates of chitosan and alginate are presented in Figure 2g and Figure 3f respectively. From linear fits we extracted the growth rates and commented on potential reasons for the differences observed. We have made the following additions to the main text:

Line 167-171

“If we assume that the initial growth phase is linear (thickness < 0.5 mm) with we obtain growth profiles of 0.83 ($r = 0.9930$) and $0.52 \text{ mm}^3 \text{ mC}^{-1}$ ($r = 0.9931$) for the high and low M_w chitosan, respectively. This means that more hydrogel is formed per unit charge for high M_w chitosan (1.6 times faster) compared to low M_w . This is likely because fewer macromolecules (bonds) are required to produce the same volume of material.”

Line 198-205

“The hydrogel thickness appears to be proportional to the cumulative amount of supplied charge density (Fig. 3f). The growth profiles (thickness per unit of charge) for gels grown at 3, 4 and 5V are not significantly different. An average growth profile of $0.17 \text{ mm}^3 \text{ mC}^{-1}$ was extracted. This result indicates that at higher potentials, the rate of H^+ generation is enhanced but the gelation mechanism remains the same. The growth profile of alginate hydrogel is 4.8 and 3 times smaller than high and low M_w chitosan hydrogels, respectively. Due to the different gelation mechanisms, a direct comparison between the growth profiles of chitosan and alginate is challenging.”

In our experiments, we fixed the electric potential (e.g. at reaction peak) and gelation time (Figures 2e, 2f, 3d and 3e). Thickness, volume and total charge density were measured. We therefore have error bars on both axis in Figures 2g and 3f. We have included additional clarification as follows (*Lines 123-124 and 180-182*):

“Thickness, volume and total charge density here are measured quantities with the error bars representing the mean \pm standard deviation ($n = 3$ independent electrodeposition experiments).”

The PEDOT/alginate hybrid presented a significantly slower growth rate. We understand it is likely because of the mechanism behind the electrochemical reactions. For the chitosan and alginate, it starts with small ion/molecule and finishes with a single bond that crosslink two macromolecules. On the other hand, the EDOT monomer electropolymerization require huge number of units to build up the PEDOT chain, requiring much more charge and is further significantly limited by diffusion of EDOT to the interface. We have added the following to the main text, *Line 269-274*:

“For the chitosan and alginate systems the electrochemical reaction generates cross-links between two macromolecules. In contrast electrodeposition of PEDOT proceeds by chain formation from small EDOT monomers. This is consistent with our observation that the PEDOT/alginate hybrid forms at a much slower rate. The electrodeposition rate may be further limited by slow diffusion of EDOT monomers in the viscous alginate solution.”

Impedance (EIS)-based gel characterization is very significant (Fig. 4e & g). The unit in the text (and Methods section) is incorrect. kW (ohm) cm ⁻² should be kW (ohm) cm ^{+ 2}

Indeed, the unit was wrong, we appreciate the opportunity to correct this. We understand the definition of resistance is electric potential divided by current, therefore the current should be normalized by area (our physical measurement), resulting in $\Omega \text{ m}^2$. The correct unit was replaced and the calculations were revised at the updated version of the manuscript in relevant panels of Figure 4, a new Supplementary Figure 4 and the Methods section (*Line 362*).

P.8, Line 137; "However, the reported reaction occurs at +1.396V, while water hydrolysis oxidation is +1.230V, being impossible to obtain a clear hydrogel without cavities in aqueous media (Supplementary Fig. 2)." It is not sufficient here from the point of the difference in standard electrode potential alone. The reaction rates of equation a, b, and c may depend on the material of the electrode. The reaction rate may change depending on whether Cl⁻ is adsorbed on the electrode or not. It is recommended to consider at least multiple conditions for Cl⁻ concentration and perform electrochemical measurement. On the other hand, fig.2d compares only with and without Cl⁻ ion. Author should show the exact concentration of Cl⁻ ion for the fig.2d, with Cl⁻ ion condition.

We agree the statement is too imperative and it is hard to be defended only based on standard electrode potentials. The standard electrode reactions give us, from thermodynamic point of view, the energy for the reaction to occur. As the reviewer points out the energy for an electrochemical reaction can drastically change depending on the metallic surface, as happens during electrocatalysis including on Gold surfaces.

However, in further support of our hypothesis we would like to mention studies that reported only AuCl_4^- and gold oxide (in form of AuO or AuOH) formation on gold electrodes in aqueous media containing chloride ions (and absence, using ClO_4^-) (DOI: 10.1016/S0022-0728(75)80004-0 and 10.1149/1.1838526). In these two studies, the concentration of chloride was varied and reaction energies were similar to the ones we studied. Furthermore, if the electrochemical reaction generates gas (in this case Cl_2

species), we would expect to see; 1) small cavities at the bottom of the hydrogel and 2) perturbations in the CV current near the reaction peak, when gas bubbles change the electrode surface in contact with solution. Based not only in the standard reduction reactions, but the additional arguments above, we hypothesize that hydrogel formation is due to AuCl_4^- instead OCl^- .

We have rewritten the paragraph to take into account the reviewer's comment (*Lines 140-147*)

“Previously it was reported that covalently crosslinked chitosan hydrogels are formed via hypochlorite (ClO^-) generated from oxidation of Cl^- to chlorine gas (Cl_2), followed by dissolution of the gas in aqueous solution^{30,32,41}. However in our system we believe gelation proceeds via tetrachloroaurate (AuCl_4^-) generation instead. This is because we were unable to form covalently crosslinked chitosan in the absence of a gold surface (i.e. on an ITO surface). Furthermore, if the electrochemical reaction generates gas (in this case Cl_2 species), we would expect to see small cavities at the bottom of the hydrogel and perturbations in the CV current near the reaction peak, when gas bubbles change the electrode surface in contact with solution (Supplementary Fig. 2).”

Please also see updated caption of Supplementary Figure 2.

We updated Figure 2d and its caption to include the concentration of the Cl^- ion. The concentration of 0.15M NaCl was selected because it was previously optimized (from 0 to 0.5M) for chitosan deposition in different study (DOI: 10.1039/c3sm27581g).

Reviewer #2 (Remarks to the Author):

The current by Da Silva et al. describes an electro-assisted printing approach using a commercial 3D printer with a potentiostat. This was done by modification of standard printing nozzles to incorporate a (Ag/AgCl) pseudoreference electrode, using in this way a three-electrode set up.

The manuscript reads well with a well-executed experimental design. The formation of covalently formed hydrogels by the 3-electrode set up is quite interesting, and represents a step-change in capability compared to the 2-electrode systems used previously. The work will therefore be of significance to those working in the field of soft neural electrodes where the connection between metal conductors and the polymer material can easily be compromised. The methodology is sound and appears to be described in enough detail to undertake replication of the work.

This manuscript has the potential to show that the author's electro-assisted printing approach could offer an all-in-one set-up for controlled polymerization of hydrogels with patterning capabilities. My only suggestion is that the authors try to place a few more lines of text into the introduction to further emphasise the novelty of this work over the 2D electrode work that they introduce. Otherwise, I think this work is well suited for publication in Nature Communications.

We thank the reviewer for reviewing our work and the positive comments.

We took the opportunity to clarify the benefits of a 3-electrode system brings over previously reported 2-electrode systems by including the following to the introduction section *Lines 66-70*.

“While two electrodes are sufficient to trigger electrochemical reactions (e.g. electrolysis), selecting which reactions occur and controlling their rate is challenging because electrode potentials are uncertain. When producing biomaterials such as hydrogels, poor control over gelation reactions may have implications on the biodegradation, mechanical, electrical or biochemical properties of the material.”

Minor change:

References should be added for this work described as: “...previous approaches that rely on 2-electrodes”

In lines 71-72, we added the following two references to the sentence in question: DOI 10.1088/1758-5090/aa6ed8 and 10.1016/j.electacta.2018.05.124

Reviewer #3 (Remarks to the Author):

Paper Review

General Comments

The manuscript titled “Electro-assisted printing of soft hydrogels via controlled electrochemical reactions” describes a study using electrochemical processes to polymerise hydrogel materials using typical materials (alginate, chitosan), and hybrid materials that also incorporate the organic conducting polymer PEDOT. The study describes and characterises the use of electrochemical control to drive the polymerisation processes of the material, incorporating with this system with a fabrication approach that allows the patterning of the hydrogel material. This is an interesting study that presents an innovative approach to polymerising and patterning hydrogel-based materials for biomedical applications. I have some specific queries/comments for the authors consideration below.

Lines 35 - 37. This sentence is confusing and is not clear. Please revise.

Lines 35 - 37. There are quite a number of self-citations from the authors when discussing the incorporation of conducting polymers in hydrogels and other types of biomaterials. However, there are also other worthy seminal works that should be included. Please include some of the major works from the field that have been omitted from the references here.

We thank the reviewer for the constructive comments and the opportunity to improve the clarity of the text. The sentences in question have been rewritten and a number of additional references were added (*Lines 34-39*).

“More recently, hydrogels have attracted attention in the field of bioelectronic interfaces because the incorporation of conductive polymers can enhance their electrical properties. A number of hybrid materials containing PEDOT, PPy, PANI and a biomaterial formed from starch, PVA, acrylic acid, polyacrylamide, alginate, decellularised extracellular matrix or others have been reported¹²⁻²¹. Such materials can form the basis of technologies for interfacing living tissues as more biocompatible and versatile alternatives to metals and plastics.”

Lines 100 – 108. The relative surface area of the counter electrode vs the working electrode will change depending on the diameter of the droplet on the surface. Did the authors identify any critical relationship between the ratio of CE:WE size and how this influenced the electrochemistry? This is particularly relevant for the characterisation of the materials using CV or EIS, as the surface area of the porous hydrogel material will (presumably) be significantly greater than the surface area of the needle counter electrode.

We thank the reviewer for pointing this out. In the nozzles we use, the inner walls are also conductive and connected to the circuit. The area of the CE was calculated and added to the following text in the discussion (*Lines 243-247*):

“In typical electrochemical cells, the area of the CE is considerably larger than that of the WE. In our setup, the area of the CE comprises not only the tip of the nozzle but also its internal walls. With an estimated area of 40.7 mm² the CE in our experiments is nearly two times larger than the footprint of alginate and chitosan droplets. Accordingly, we did not observe any side reaction or obstruction of the nozzle during gelation.”

Also, given polymerisation can take some time (several minutes), did the authors have any issues with the needle becoming blocked through the polymerisation of material within the needle itself?

We did not observe blocking of printing nozzles due to polymerised material inside, because the polymerisation reactions were taking place at the WE (gold surface).

Line 143. Correct spelling to 'stabilized'.

Spelling was corrected.

Line 221 – 223. “Beyond 6 minutes, the polymerisation current begins to decrease probably because the electrodeposited material has covered the WE surface and starts to grow in thickness.” During this polymerisation process is material being actively released through the needle, or is it stagnant? If the flow of material is stagnant, the dropping off of the potential may be that the concentration of monomer in the polymerisation medium is becoming exhausted, and is not being replenished through introduction of new material through the needle, thus polymerisation rate is decreasing.

Indeed, we agree the point raised is very relevant for the discussion. In the experiment reported in Figure 4c, the solution was stagnant. Indeed, the depletion of monomers play an important role in the decreasing polymerization rate. We included the following sentence in the revised manuscript (*Lines 231-233 and 269-274*):

“Beyond 6 minutes, as the electrodeposited material covered the WE surface, the polymerisation current (Fig. 4c) begins to decrease which is likely due to depletion of monomers in the electrolyte.”

“For the chitosan and alginate systems the electrochemical reaction generates cross-links between two macromolecules. In contrast electrodeposition of PEDOT proceeds by chain formation from small EDOT monomers. This is consistent with our observation that the PEDOT/alginate hybrid forms at a much slower rate. The electrodeposition rate may be further limited by slow diffusion of EDOT monomers in the viscous alginate solution.”

Line 224. Add bracket.

This was corrected.

REVIEWERS' COMMENTS

Reviewer #1 (Remarks to the Author):

Authors replied almost all the questions from the referees. I recommend publication of the article as is.

Reviewer #2 (Remarks to the Author):

The authors have addressed my comments in full.

Reviewer #3 (Remarks to the Author):

The authors have studiously addressed all of the reviewer comments, that were focused on several areas of the manuscript. I am happy with the authors comments and responses to each of the reviewer comments, and believe they have appropriately amended the manuscript.